# NovoBench: Benchmarking Deep Learning-based *De Novo* Peptide Sequencing Methods in Proteomics

**Jingbo Zhou**[1,2]**, Shaorong Chen**[1,2*]**, Jun Xia**[1,2†]**, Sizhe Liu**[3]**, Tianze Ling**[4]
**Wenjie Du**[2]**, Yue Liu**[2]**, Jianwei Yin**[1]**, Stan Z. Li**[2†]
[1]Zhejiang University, [2]Westlake University,
[3]University of Southern California [4]Tsinghua Univerisity
{zhoujingbo, xiajun, stan.zq.li}@westlake.edu.cn

## Abstract

Tandem mass spectrometry has played a pivotal role in advancing proteomics, enabling the high-throughput analysis of protein composition in biological tissues. Many deep learning methods have been developed for *de novo* peptide sequencing task, i.e., predicting the peptide sequence for the observed mass spectrum. However, two key challenges seriously hinder the further advancement of this important task. Firstly, since there is no consensus for the evaluation datasets, the empirical results in different research papers are often not comparable, leading to unfair comparison. Secondly, the current methods are usually limited to amino acid-level or peptide-level precision and recall metrics. In this work, we present the first unified benchmark NovoBench for *de novo* peptide sequencing, which comprises diverse mass spectrum data, integrated models, and comprehensive evaluation metrics. Recent impressive methods, including DeepNovo, PointNovo, Casanovo, InstaNovo, AdaNovo and $\pi$-HelixNovo are integrated into our framework. In addition to amino acid-level and peptide-level precision and recall, we evaluate the models' performance in terms of identifying post-tranlational modifications (PTMs), efficiency and robustness to peptide length, noise peaks and missing fragment ratio, which are important influencing factors while seldom be considered. Leveraging this benchmark, we conduct a large-scale study of current methods, report many insightful findings that open up new possibilities for future development. The code is available at `https://github.com/Westlake-OmicsAI/NovoBench`.

## 1 Introduction

Proteomics, the study of proteins within biological systems, relies heavily on mass spectrometry for protein identification [1]. Traditional methods use existing databases to match observed peptide fragments with known sequences. However, these methods may miss novel or modified peptides not present in the databases [23, 22, 29]. *De novo* peptide sequencing offers a solution by directly annotating mass spectra to reconstruct peptide sequences without relying on databases. By circumventing the need for predefined databases, *de novo* sequencing enables researchers to uncover new peptides and investigate post-translational modifications (PTMs), contributing to a deeper understanding of cellular processes [42] and disease mechanisms [24]. Here, PTMs refers to chemically modified version of 20 naturally-occurring amino acids, influencing many critical biomolecular processes including central enzyme activities, protein turnover, and DNA repair [33, 6, 30]. Deep learning has been widely used in the *de novo* peptide sequencing, where researchers "translate" the observed mass spectrum to peptide sequence using the encoder-decoder architectures [40, 32, 52, 25, 49, 21, 54].

---

*Equal contribution.
†Corresponding author.

38th Conference on Neural Information Processing Systems (NeurIPS 2024) Track on Datasets and Benchmarks.

Despite the remarkable success, there exists three key challenges that seriously hinder the further development of deep learning-based *de novo* peptide sequencing:

- **The datasets for fair evaluation**. Because the Peptide-Spectrum Matches (PSMs) for training and evaluation are easily accessible in ProteomeXchange [46], researchers may download different parts for the evaluation of their respective models in *de novo* peptide sequencing. For example, DeepNovo [40] and PointNovo [32] report the performance on the seven-species dataset, while InstaNovo [7] evaluate the performance on the other data collected by themselves. Furthermore, there are multiple versions of the dataset available, and the datasets used by the models with the same name are actually different. For example, PointNovo and CasaNovo use different versions of the Nine-species dataset (MassIVE dataset identifier: MSV000090982, MSV000081382). The inconsistency of the datasets used would potentially obscure the true progress in the field.

- **The metrics for comprehensive evaluation**. Although previous works in *de novo* peptide sequencing share the metrics of peptide-level or amino acids-level precision and recall, they fail to evaluate some important abilities of the models. For example, PTMs play an essential role in elucidating protein functions, however, we find that current methods struggle to identify them compared to naturally-observing amnio acids. Hence, it is necessary to establish some metrics to evaluate the models' abilities in identifying PTMs. Moreover, we observe that some models are computationally extensive, hindering the practical deployment of these tools and necessitating the metrics for training and inference efficiency.

- **The robustness to important influencing factors**. Intuitively, longer peptide sequences and a higher noise peaks ratio are expected to degrade the performance of various models. However, the extent to which these factors affect different models remains unknown. This underscores the importance of selecting the appropriate model for specific scenarios. Additionally, another crucial factor is the missing fragmentation ratio. Previous studies[27] have indicated that missing fragmentation during peptide sequencing results in insufficient information for determining peptide sequences in those regions, increasing the likelihood of errors in the resulting peptide identifications. However, few previous works have evaluated the impact of this factor on model performance.

To solve these challenges, we develop the first deep learning-based *de novo* sequencing benchmark that supports unified, reproducible, and efficient evaluations. We comprehensively integrated deep learning-based *de novo* peptide sequencing models including DeepNovo[40], PointNovo[32], CasaNovo[52], InstaNovo[7], AdaNovo[49], and π-HelixNovo[51] into our framework. Additionally, in addition to amino acids-level or peptide-level precision and recall, we take new metrics including PTMs precision and efficiency. Based on this benchmark, we conduct extensive experiments to compare different models in a fair fashion. We also investigate the models' robustness to the three key influencing factors, providing guidance for model selection according to specific applications.

## 2    Background and Task Definition

Protein identification is a central objective in proteomics-related analyses. The liquid chromatography-tandem mass spectrometry (LC-MS/MS) technique is widely utilized for both the quantitative and qualitative analysis of proteins. In the typical protein identification workflow of shotgun proteomics, proteins are first digested by enzymes to generate a mixture of peptides [47].

These peptides are then separated using liquid chromatography. Each charged peptide is analyzed by a mass spectrometer, producing first scan (MS1) spectra that show the mass-to-charge (*m/z*) ratios of the intact peptides. Subsequently, the peptides are fragmented in the mass spectrometer, yielding second scan (MS2) spectra, which consist of multiple peaks, each defined by an *m/z* value and an intensity value. A crucial part of protein identification is peptide sequencing, which involves predicting the peptide sequence from the observed MS2 spectrum and precursors (the mass and charge of the peptide). Unlike database search-based peptide sequencing methods, *de novo* peptide sequencing relies solely on the MS2 spectrum information without using any pre-constructed databases. Finally, the complete protein sequence can be deduced using assembly methods [20].

The precise task definition of *de novo* peptide sequencing is the process of reconstructing a peptide's amino acid sequence directly from an observed MS2 spectrum, along with the precursor charge and

mass. The MS2 spectrum can be denoted as as $\mathbf{x} = \{(m_i, I_i)\}_{i=1}^{M}$, where each peak $(m_i, I_i)$ forms a 2-tuple representing the *m/z* and intensity value, and $M$ is the number of peaks that can be varied across different mass spectra. The precursor charge and mass are represented as an integer indicating the charge and a floating-point number indicating the mass, respectively. Additionally, we represent the peptide sequence which we want to identify as $\mathbf{y} = \{(y_1, y_2, \ldots, y_N)\}$, where $y_i$ is the type of the $i$-th amino acid, $N$ is the peptide length and can be varied across different peptides.

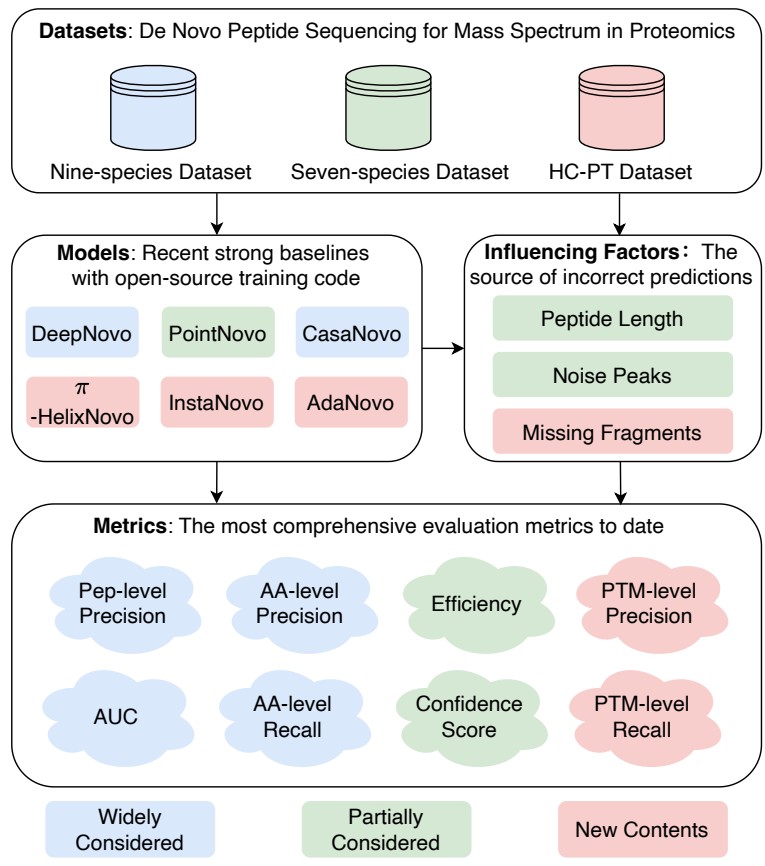

Figure 1: The overview of the NovoBench benchmark. The benchmark is organized incrementally from datasets to models, to metrics. We color contents in green and blue that are widely and partially considered by previous studies, respectively. Newly introduced contents are colored in pink.

## 3 Datasets

In this paper, we selected three representative datasets: Seven-species, Nine-species, and HC-PT. These datasets, varying in size, exhibit diversity in spectrum resolution and peptide sources, enabling a more comprehensive and accurate evaluation of current *de novo* peptide sequencing models. The detailed properties of the datasets are shown in Table 1.

Table 1: The datasets statistics of NovoBench.

| Dataset | precusor m/z | precusor charge | *Avg.* peaks num. | intensity | peptide len. | PTM class | min *m/z* | max *m/z* | train/valid/test num. |
|---|---|---|---|---|---|---|---|---|---|
| Seven-species | 719.07 | 2.42 | 466.05 | 956.17 | 15.79 | 3 | 70.17 | 3997.66 | 317,009 / 17,740 / 17,094 |
| Nine-species | 679.68 | 2.47 | 134.91 | 175082.65 | 15.01 | 3 | 53.03 | 35932.63 | 499,402 / 28,572 / 27,142 |
| HC-PT | 635.32 | 2.31 | 184.21 | 143363.17 | 12.53 | 1 | 99.99 | 1999.99 | 213,284 / 25,718 / 26,536 |

**Seven-species Dataset.** The Seven-species dataset contains low-resolution mass spectrum and their peptide labels from 7 different species. The previous work DeepNovo [40] has evaluated its performance on these datasets with the leave-one-out method, i.e., training the model on 6 species and testing on the left one species, to mimic the real-world challenging cases where we have to identify the never-before-seen peptide sequences for the observed mass spectrum. In this paper, we conducted testing on the yeast species and training on the remaining 6 species.

**Nine-species Dataset.** The Nine-species dataset is the most widely-used dataset by previous works such as DeepNovo [40], PointNovo [32], and Casanovo [52], which contains high-resolution mass spectrum and their peptide labels from 9 different species. We adopt the Nine-species dataset used by the original publication of DeepNovo (MassIVE dataset identifier: MSV000081382) for benchmarking. Similar to Seven-species dataset, we train models on 8 species and evaluate the left yeast species. Additionally, these datasets contain 3 PTMs (oxidation of methionine, deamidation of asparagine or glutamine), enabling the fair evaluation of various models' performance in terms of identifying PTMs.

**HC-PT Dataset.** The HC-PT dataset, as detailed in the InstaNovo paper[7], includes synthetic tryptic peptides that span all canonical human proteins and isoforms. It also encompasses peptides generated by alternative proteases and HLA peptides. The key feature of the HC-PT dataset is its high-resolution spectrum for human-origin peptides, and the peptide labels are derived from the high-confidence search results of MaxQuant[41].

## 4 Baseline models

Due to advancements in deep learning, many neural network-based models have been developed for the *de novo* peptide sequencing task. These methods can be broadly classified into two categories: those based on traditional deep learning techniques such as CNN and LSTM and those based on the Transformer architecture. We selected six representative models from these two categories as benchmark models for testing.

**Traditional Deep Learning Methods.** DeepNovo, as the pioneering model, integrates deep learning into de novo peptide sequencing. It discretizes the input mass spectrum into vector representation using predefined bin sizes and channels them into ion-CNN for effective processing. The resulting output from ion-CNN undergoes further feature extraction through LSTM, and the combined outputs from both modules are fused to predict the succeeding amino acid in the sequence. To ensure precise decoding, the algorithm employs a knapsack algorithm that constrains the predicted peptide mass and the observed precursor mass within a specific range. PointNovo enhances the discretization approach of DeepNovo by utilizing sets of (*m/z*, intensity) pairs as the input to the model. It adopts an architecture similar to PointNet for accurate prediction of the next amino acid.

**Transformer Models.** CasaNovo employs the Transformer as both the encoder and decoder. The encoder takes variable-length sets of (*m/z*, intensity) pairs as input, while the precursor mass and charge are provided as additional inputs to the decoder, which iteratively predicts the next amino acid. InstaNovo uses a similar architecture to CasaNovo but adding a precursor encoder. The output of the precursor encoder and the spectrum encoder are concatenated into the decoder for decoding. AdaNovo, a novel framework for addressing PTM (post-translational modification) identification problems, adds a peptide decoder to calculate the spectrum-peptide and amino acid mutual information for adaptive model training. $\pi$-HelixNovo addresses the issue of missing ions in MS2 by introducing a complementary spectrum as a supplement to each experimental spectrum.

## 5 Metrics

In this section, we introduce metrics that will be used for *de novo* peptide sequencing evaluation, including precision and recall at multiple levels (peptide level, amino acid level, and amino acid with PTM level), confidence scores, area under the precision-recall curve and efficiency.

Previous works[40, 32, 52, 25, 49, 21] mainly focus on improving amino acid-level or peptide-level precision and recall, while ignoring other metrics. However, we argue that recovery is not the only important metric for *de novo* peptide sequencing evaluation. Other metrics introduced follows are also crucial for comprehensively revisiting current approaches. For example, PTMs play an essential

role in elucidating protein functions but have lower accuracy than common amino acids. Therefore, this paper introduces PTM level metrics.

In addition to metrics related to prediction accuracy, practical applications of models often place significant emphasis on the model's efficiency and the confidence scores of its predictions. This paper also incorporates these metrics into the evaluation.

**Amino acid-level Precision and Recall.** We first calculate the number of matched amino acid predictions, $N_{\text{match}}^{aa}$, which are defined as predicted amino acids that (1) exhibit a mass difference of $< 0.1\text{Da}$ from the corresponding ground truth amino acid and (2) have either a prefix or suffix with a mass difference of $\leq 0.5\text{Da}$ from the corresponding amino acid sequence in the ground truth peptide. Amino acid-level precision and recall is then defined as $N_{\text{match}}^{aa}/N_{\text{pred}}^{aa}$ and $N_{\text{match}}^{aa}/N_{\text{truth}}^{aa}$, where $N_{\text{pred}}^{aa}$ and $N_{\text{truth}}^{aa}$ represent the number of predicted amino acids in predicted peptide sequences and ground truth peptide sequences, respectively.

**Peptide-level Precision.** Compared with amino acid-level performance, peptide-level performance measures are the primary quantifier of the model's practical utility because the goal of *de novo* peptide sequencing is to assign a complete peptide sequence to each spectrum. For peptide prediction, a predicted peptide is deemed a correct match only if all of its amino acids are matched according to the definition mentioned above. In a collection of $N_{\text{all}}^{p}$ spectra, if a model accurately predicts $N_{\text{match}}^{p}$ peptides, the peptide-level precision are defined as $N_{\text{match}}^{p}/N_{\text{all}}^{p}$.

**PTM-level Precision and Recall.** PTMs play an essential role in elucidating protein functions. However, current models show significantly lower accuracy for amino acids with PTMs compared to other common amino acids, and this metric is also less frequently used as an evaluation. Similar to amino acid-level metrics, PTMs identification precision and recall can be formulated as $N_{\text{match}}^{ptm}/N_{\text{pred}}^{ptm}$ and $N_{\text{match}}^{ptm}/N_{\text{orig}}^{ptm}$, where $N_{\text{match}}^{ptm}$, $N_{\text{pred}}^{ptm}$ and $N_{\text{orig}}^{ptm}$ denote the number of matched PTMs, predicted amino acids with PTMs and PTMs in ground truth peptide sequence, respectively.

**Confidence.** Calculating precision and recall requires access to the reference sequence, which is not always available in practice. When the ground-truth sequence is unknown, measuring and ranking the quality of the predicted sequence becomes more challenging. We introduce the confidence metric to address this problem, which is the average predictive probability of predicted amino acids, defined as:

$$\texttt{Conf} = \frac{1}{N} \sum_{i=1}^{n} p\left(\hat{y}_i\right), \tag{1}$$

We adopt the mean score of all amino acids as a peptide-level confidence score, where $N$ is the peptide length, and $p\left(\hat{y}_i\right)$ is the predicted probability of the amino acid type $\hat{y}_i$ output by the model.

**Area under the precision-recall curve (AUC).** Given the availability of peptide recall, precision, and confidence scores, it is reasonable to draw precision-recall curves and use the area under the curve (AUC) as a summary of *de novo* sequencing accuracy. To achieve this, sort the prediction results of each model from high to low based on confidence scores. Starting from the highest confidence prediction, accumulate the model's peptide recall and precision values. These cumulative values are then used as the x and y coordinates, respectively, for the points on the precision-recall curve. The AUC of the precision-recall curve is an effective metric for evaluating classification models on imbalanced datasets, providing a comprehensive assessment of the model's performance under different confidence scores.

**Efficiency.** Efficiency measures the computational resources and time required to generate the peptide sequence. This study reports the training time, inference time, and model parameters of different methods over the standard benchmarks. While efficiency may not be a crucial problem compared to precision and recall, it is a useful metric for assessing the model's scalability and practicality.

## 6 Influencing Factors

In this section, we investigate the influence of various factors on performance of *de novo* peptide sequence models to evaluate the robustness of all models against varying degrees of missing fragmentation, noise peaks and peptide lengths.

**Peptide Length.** We first analyze the impact of peptide length. Peptide length is defined as the number of amino acids in a peptide chain. Generally, the longer the peptide length, the more challenging it is for the model to make accurate predictions.

**Noise Peaks.** Noise is another crucial factor contributing to the complexity of resolving spectra. It can originate from various sources, such as white noise from the instruments, chemical contaminants, and unexpected fragment ions. To evaluate the impact of noise, we reconstruct the theoretical spectrum considering several types of ions based on the ground-truth peptide sequence. Peaks from the experimental spectrum that appear in the theoretical spectrum within the error tolerance are considered signal peaks, while others are classified as noise peaks. We then calculate the ratio of noise peaks to signal peaks in the spectrum, termed the Noise Signal Ratio (NSR), to measure the noise degree of the spectrum.

**Missing Fragmentation Ratio.** To further explore the influence of missing fragmentation, we assess the impact of varying degrees of missing fragmentation on the performance of all models. Given that recall and precision calculations are related to peptide length, we define the Missing Fragmentation Ratio (MFR). MFR quantifies the extent of fragmentation information loss in the spectrum and is calculated as the number of missing fragmentations divided by the number of candidate fragmentation sites along the peptide.

# 7 Results and Analysis

## 7.1 Experimental Settings

The model hyperparameters used in this paper are consistent with those in the original papers. We set the batch size to 32 and train the models for 30 epochs. DeepNovo and PointNovo were validated every 3000 steps, while the other models were validated every 50000 steps. The model with the lowest validation loss was selected for testing. **We used the official pre-trained Instanovo model for testing on the Nine-species and HC-PT datasets, while the remaining models were trained from scratch on three datasets.** All experimental results were obtained using a Nvidia A100 GPU (80GB). Please refer to the supplementary material for the detailed settings of each model.

## 7.2 Main Results

Table 2: Empirical comparison of *de novo* sequencing models using amino acid-level and peptide-level metrics. The best and the second best are highlighted with **bold** and underline, respectively.

| Method | Amino acid-level performance | | | | | | Peptide-level performance | | | | | |
| | Seven-species | | Nine-species | | HC-PT | | Seven-species | | Nine-species | | HC-PT | |
| | Prec. | Recall | Prec. | Recall | Prec. | Recall | Prec. | AUC | Prec. | AUC | Prec. | AUC |
|---|---|---|---|---|---|---|---|---|---|---|---|---|
| DeepNovo | **0.492** | 0.433 | 0.696 | 0.638 | 0.531 | 0.534 | 0.204 | 0.136 | 0.428 | 0.376 | 0.313 | 0.255 |
| PointNovo | 0.196 | 0.169 | 0.740 | 0.671 | 0.623 | 0.622 | 0.022 | 0.007 | 0.480 | 0.436 | 0.419 | 0.373 |
| CasaNovo | 0.322 | 0.327 | 0.697 | 0.696 | 0.442 | 0.453 | 0.119 | 0.084 | 0.481 | 0.439 | 0.211 | 0.177 |
| InstaNovo | - | - | 0.736 | 0.703 | **0.714** | **0.712** | - | - | **0.532** | **0.495** | **0.572** | **0.549** |
| AdaNovo | 0.379 | 0.385 | 0.698 | 0.709 | 0.442 | 0.451 | 0.174 | 0.135 | 0.505 | 0.469 | 0.212 | 0.178 |
| $\pi$-HelixNovo | 0.481 | **0.472** | **0.765** | **0.758** | 0.588 | 0.582 | **0.234** | **0.173** | 0.517 | 0.453 | 0.356 | 0.318 |

*De novo* **peptide sequencing.** As can be observed from Table 2, de novo sequencing models reveals varied performance at both the amino acid and peptide levels across different datasets. At the amino acid level, $\pi$-HelixNovo stands out with the high precision and recall across the Seven-species and Nine-species datasets, achieving precision and recall values of 0.481 and 0.472, and 0.765 and 0.758, respectively. InstaNovo achieved the highest precision (0.714) and recall (0.712) on the HC-PT dataset. DeepNovo exhibits the highest precision (0.492) on the Seven-species dataset but shows less impressive performance on the other datasets. PointNovo demonstrated the second-highest precision and recall (0.623 and 0.622) on the HC-PT dataset. However, compared to its performance on the high-resolution datasets Nine-species and HC-PT, the model did not perform as well on the low-resolution Seven-species dataset. CasaNovo shows consistent but average performance across all datasets. AdaNovo performs well on the Nine-species dataset, with a recall of 0.709, but is otherwise unremarkable.

At the peptide level, InstaNovo demonstrates superior performance on the Nine-species and HC-PT datasets, with the highest precision and AUC values. $\pi$-HelixNovo achieved the best precision and AUC value on the Seven-species dataset. In contrast, DeepNovo and CasaNovo show moderate performance, with DeepNovo slightly edging out in precision for the Seven-species dataset (0.204). PointNovo excels on the HC-PT dataset, achieving the second-highest precision (0.419) and AUC (0.373) among all models. AdaNovo achieves the second-highest AUC (0.469) on Nine-species dataset. Regarding confidence score, as shown in Table 4, $\pi$-HelixNovo has the highest confidence score, while InstaNovo has the lowest. Additionally, the confidence scores of Transformer-based models are notably higher compared to traditional methods.

**Identifying PTMs.** The results in Table 3 show that DeepNovo achieves the highest PTM recall for the Seven-species dataset (0.373) and performs well on both the Nine-species and HC-PT datasets. PointNovo excels in the HC-PT dataset, attaining the second-highest PTM recall (0.740) and precision (0.676). CasaNovo demonstrates high PTM precision on the Nine-species dataset (0.706). InstaNovo achieves the highest performance on HC-PT dataset. AdaNovo performs competitively, achieving the second-highest PTM recall for the nine-species dataset (0.570) and a PTM precision of 0.448 for the Seven-species dataset. $\pi$-HelixNovo exhibits consistent performance across all datasets, securing the top PTM recall for the Nine-species dataset (0.598) and achieving notable precision scores.

Overall, $\pi$-HelixNovo consistently performs well across all datasets and metrics, demonstrating the effectiveness of complementary spectrum in *de novo* peptide sequencing task. InstaNovo demonstrates strong performance on both the Nine-species and HC-PT datasets, highlighting its robust capabilities after pre-training on large-scale, high-resolution mass spectrometry data. Additionally, traditional deep learning-based methods remain highly competitive, and designing a better model architecture is still worth exploring.

Table 3: Empirical comparison of *de novo* sequencing models in terms of identifying PTMs. The best and the second best are highlighted with **bold** and underline, respectively.

| Method | PTM Recall | | | PTM Prec. | | |
|---|---|---|---|---|---|---|
| | Seven-species | Nine-species | HC-PT | Seven-species | Nine-species | HC-PT |
| DeepNovo | **0.373** | 0.529 | 0.615 | 0.391 | 0.576 | 0.626 |
| PointNovo | 0.094 | 0.546 | 0.740 | 0.117 | 0.629 | 0.676 |
| CasaNovo | 0.251 | 0.566 | 0.460 | 0.360 | **0.706** | 0.501 |
| InstaNovo | - | 0.550 | **0.829** | - | 0.674 | **0.754** |
| AdaNovo | 0.321 | 0.570 | 0.482 | 0.448 | 0.652 | 0.552 |
| $\pi$-HelixNovo | 0.366 | **0.598** | 0.667 | **0.473** | 0.680 | 0.568 |

Table 4: Confidence score of different models on various datasets. The symbol $*$ identifies the pre-trained model. The best and the second best are highlighted with **bold** and underline, respectively.

| Model | Seven-species | Nine-species | HC-PT | *Avg.* |
|---|---|---|---|---|
| Deepnovo | 0.515 | 0.697 | 0.503 | 0.572 |
| Pointnovo | 0.264 | 0.735 | 0.620 | 0.540 |
| CasaNovo | 0.760 | 0.905 | 0.751 | 0.805 |
| AdaNovo | 0.798 | 0.914 | 0.761 | 0.824 |
| InstaNovo | - | 0.507 | 0.565 | 0.536 |
| $\pi$-HelixNovo | **0.846** | **0.935** | **0.843** | **0.875** |

## 7.3 Influencing Factors

In this section, we evaluate the robustness of all models under varying degrees of three influencing factors: missing fragmentation, noise, and peptide lengths.

**Peptide Length.** We first analyze the impact of peptide length, defined as the number of amino acids in a peptide. For the nine-species datasets, peptides lengths are highly variable. Although the mode of peptide length is 12, there are still many peptides with lengths exceeding 25. Generally, the longer the peptide length, the more challenging it becomes for the model to accurately predict, resulting in lower peptide-level accuracy. However, the performance of models remains relatively stable when the peptide length exceeds a certain threshold. For instance, when the peptide length exceeds 14, the precision of five models (excluding Instanovo) fluctuates rather than monotonically decreasing.

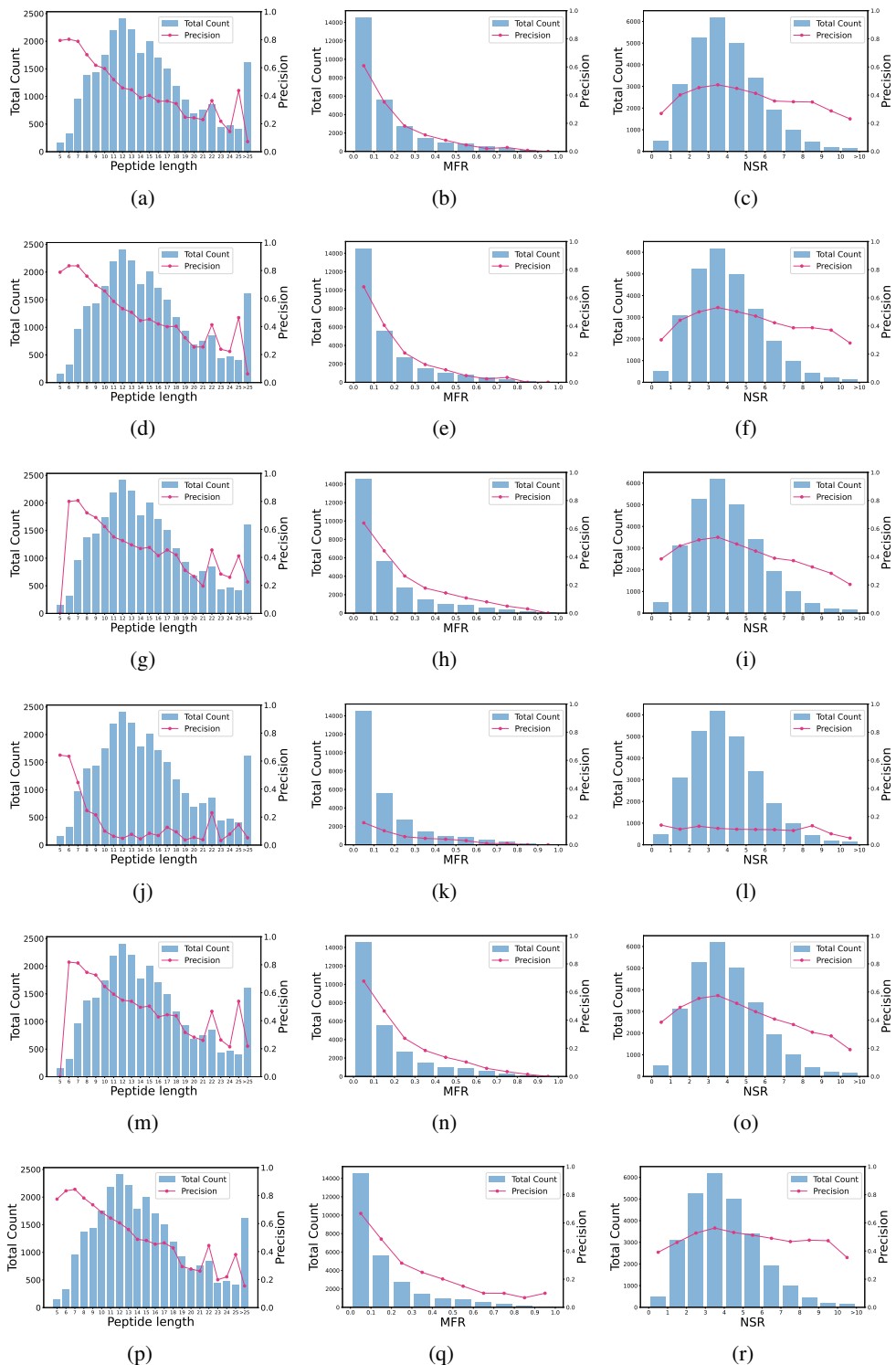

Figure 2: Peptide-level precision curves of the benchmarking models under the influence of three factors on Nine-species dataset. The first row to the sixth row correspond to the models DeepNovo, PointNovo, CasaNovo, InstaNovo, AdaNovo, $\pi$-HelixNovo, and the first column to the third column correspond to the three influencing factors Peptide length, missing fragmentation ratio (MFR) and noise peaks (NSR). More results of the other two datasets can be found in the appendix.

Notably, the precision for peptides of length 5 is slightly lower or comparable to those of lengths 6 and 7 for these five models, which may be due to the fewer training data for peptides of length 5. In contrast, the Instanovo model exhibits poor robustness to peptide length, with performance declining sharply as peptide length increases.

**Missing Fragmentation Ratio.** The Missing Fragmentation Ratio (MFR) measures the degree of missing fragmentation by quantifying the portion of fragmentation information lost in the spectrum. Statistical results show that the majority (over 50%) of the mass spectra in the nine-species datasets have an MFR in the range of [0.0, 0.1), suggesting a high level of reliability in the training data. The results demonstrate a significant decline in peptide precision as the Missing Fragmentation Ratio (MFR) increases. This trend is expected, as the recall of amino acids is low in regions with missing fragmentations, leading to the failure of predicting the entire peptide if any single amino acid is inaccurately predicted. We can conclude that nearly all *de novo* peptide sequencing models are highly affected by the degree of missing fragmentation.

**Noise Peaks.** The Noise Signal Ratio (NSR) is calculated as the ratio of noise peaks to signal peaks in the spectrum to measure the noise profile. From the NSR analysis, it is evident that in the vast majority of mass spectra, the number of noise peaks significantly exceeds the number of signal peaks (NSR > 1), indicating the widespread presence of noise information in the spectra. Generally, the performance is expected to degrade as NSR increases. However, we observe an unexpected initial increasing trend. This can be attributed to the data distribution: spectra with smaller NSR tend to have more missing fragmentations. As NSR increases, the average number of missing fragmentations decreases to a relatively stable point, leading to an initial improvement in performance. In summary, as noise increases (indicated by a rising NSR), the model's performance initially improves and then declines. However, compared to peptide length and missing fragmentation ratio, noise peaks have a relatively smaller impact on model performance.

## 7.4 Computational Efficiency

Table 5: Computational efficiency comparison of various models on the same device. The training time and inference time here refer to the averaged time over a batch.

| Model | Training Time (s) | | | Inference Time (s) | | | Trainable Params (M) |
|---|---|---|---|---|---|---|---|
| | Seven-species | Nine-species | HC-PT | Seven-species | Nine-species | HC-PT | All dataset |
| DeepNovo | 0.31 | 0.38 | 0.30 | 0.04 | 0.07 | 0.02 | 8.63 |
| PointNovo | 0.34 | 0.31 | 0.28 | 0.25 | 0.24 | 0.22 | 4.78 |
| CasaNovo | 0.36 | 0.33 | 0.32 | 0.27 | 0.28 | 0.26 | 47.3 |
| InstaNovo | - | - | - | - | 0.39 | 0.37 | 92.3 |
| AdaNovo | 1.16 | 1.07 | 0.96 | 1.48 | 1.50 | 1.46 | 66.3 |
| $\pi$-HelixNovo | 0.56 | 0.35 | 0.41 | 0.30 | 0.28 | 0.17 | 47.3 |

In this section, we compares the efficiency of various *de novo* peptide sequencing models in terms of training time, inference time, and the number of trainable parameters across three datasets. We set the batch size as 32 and recorded the time required for each model to train and infer a single step over a batch on different datasets. Additionally, we documented the number of trainable parameters for each model. From Table 5, we can observe that the parameter counts for DeepNovo and PointNovo are lower compared to models based on the Transformer architecture. Despite CasaNovo having nearly ten times the parameters of PointNovo, its training and inference time are comparable. InstaNovo significantly outperforms other models in training parameters. AdaNovo, which includes an additional peptide decoder compared to CasaNovo, has the highest training time and inference time. $\pi$-HelixNovo modifies the input spectrum, resulting in similar levels for all three metrics as CasaNovo.

## 8 Conclusion

Although *De novo* peptide sequencing has achieved remarkable advancement, the lack of thorough comparisons across diverse datasets, metrics and influencing factors hinders the progress toward practical applications. To address this issue, we propose NovoBench, which consists of diverse datasets, models, and metrics and provides a comprehensive view of deep learning-based *de novo* peptide sequencing. In the future, We plan to build an automated end-to-end computational proteomics pipeline that simplifies and standardizes the process of PSMs data loading, experimental setup, and

model evaluation for both *de novo* peptide sequencing and theoretical mass spectrum prediction. This will promote the scientific research and practical application of computational proteomics.

## 9 Limitation and Future Work

Given that Data-Independent Acquisition (DIA) data differs from Data-Dependent Acquisition (DDA) data and that there are currently fewer models [55, 56] available for DIA, the availability of open-source DIA models is limited. Furthermore, de novo peptide sequencing models have already been widely applied to DDA mass spectrometry data, so our work is currently limited to DDA data. In the future, we will support DIA mass spectrometry data and DIA-specific models.

## 10 Acknowledgements

This work was supported by National Science and Technology Major Project (No. 2022ZD0115101), National Natural Science Foundation of China Project (No. 623B2086, No. U21A20427), Project (No. WU2022A009) from the Center of Synthetic Biology and Integrated Bioengineering of Westlake University and Integrated Bioengineering of Westlake University and Project (No. WU2023C019) from the Westlake University Industries of the Future Research Funding.

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

# APPENDIX

## A. Influencing Factors

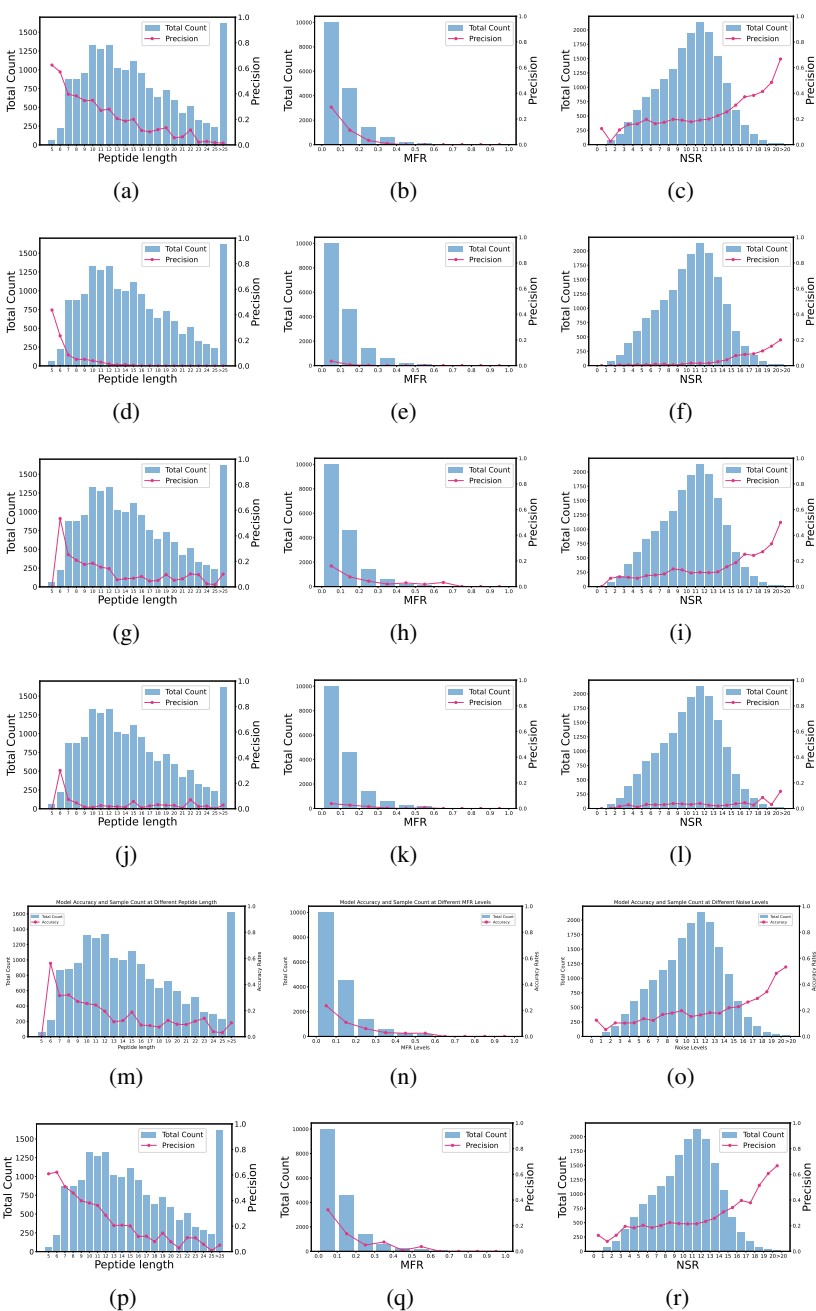

Figure 3: Peptide-level precision curves of the benchmarking models under the influence of three factors on Seven-species dataset. The first row to the sixth row correspond to the models DeepNovo, PointNovo, CasaNovo, InstaNovo, AdaNovo, $\pi$-HelixNovo, and the first column to the third column correspond to the three influencing factors Peptide length, missing fragmentation ratio (MFR) and noise peaks (NSR).

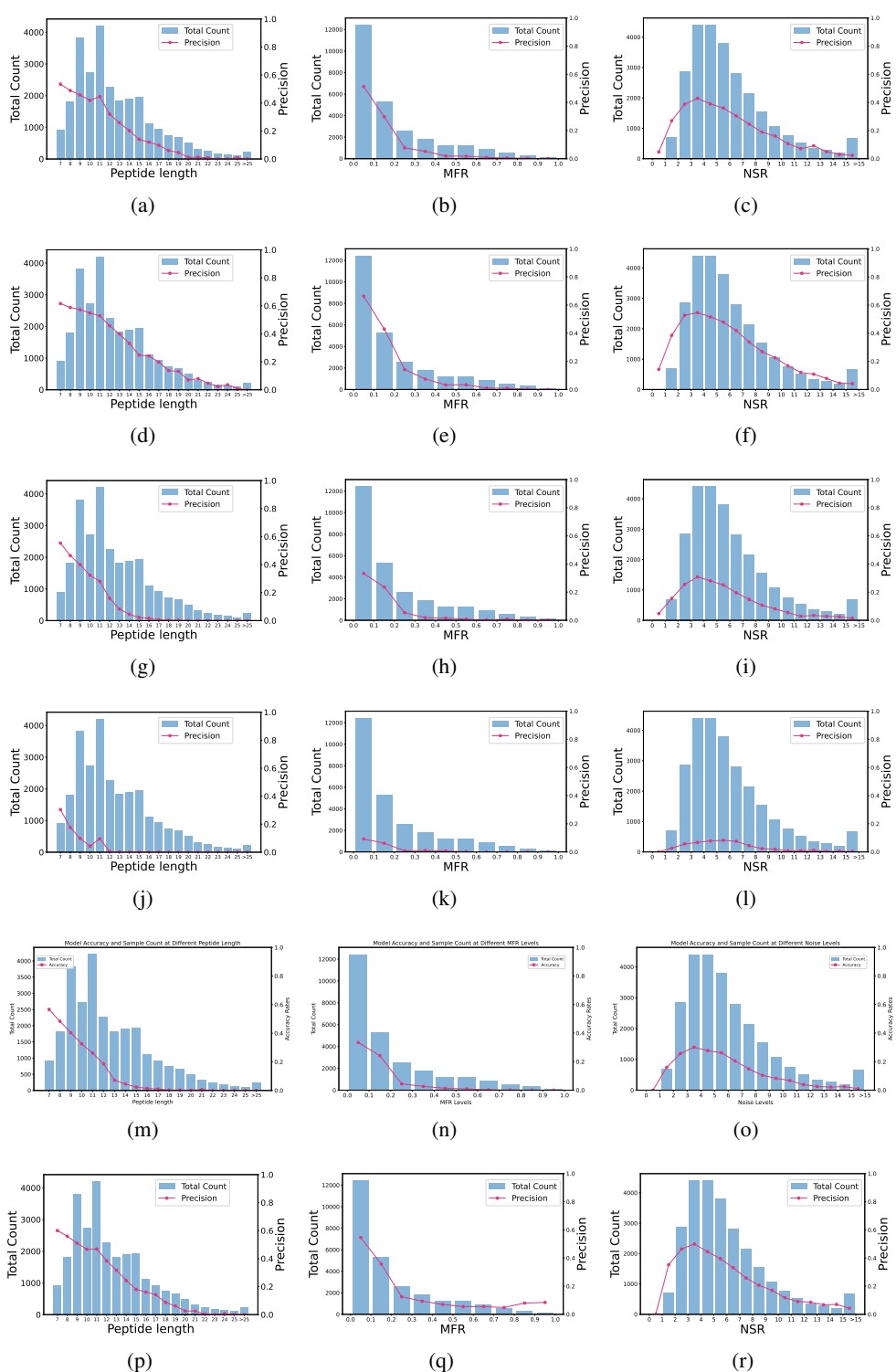

Figure 4: Peptide-level precision curves of the benchmarking models under the influence of three factors on HC-PT dataset. The first row to the sixth row correspond to the models DeepNovo, PointNovo, CasaNovo, InstaNovo, AdaNovo, $\pi$-HelixNovo, and the first column to the third column correspond to the three influencing factors Peptide length, missing fragmentation ratio (MFR) and noise peaks (NSR).

