# OpenReview forum: "NovoBench: Benchmarking Deep Learning-based \emph{De Novo} Sequencing Methods in Proteomics"
_NeurIPS.cc/2024/Datasets_and_Benchmarks_Track — NeurIPS 2024 Track Datasets and Benchmarks Poster_

### Official Review · Reviewer_9E2x · 2024-06-18
**Excellent preparation for data-track with data assessments, curation, user-friendliness, future roadmap for a dataset in important topic**

**Rating:** 7
**Confidence:** 4
**Correctness:** I did not detect any technical issues.
**Clarity:** Yes, just minor grammar issues as below.

**Review:**

I do not have any major issues with the work, the methods and analyses are clear, well-documented and straightforward. The topic is timely and important.

The limitations section is weak however, the authors should improve discussion about the weakness of current feature lists, models, data quality or availability, etc. There is basically no limitation stated in the final section though fair limitations are scattered throughout the text in other places.

Given how much the MFR degrades model performance, can the authors also improve discussion of how theorists and/or experimentalists are trying to improve that issue in recent work?

**Strengths:**

The problem the authors are tackling is high impact and the manuscript writing and methods are high quality.

**Additional Feedback:**

It was a pleasure to read this paper, it sets a strong example of how a data-track paper should be written.

**Documentation:**

The github link works and the in-text documentation is thorough and helpful.

**Ethics:**

No.

**Limitations:**

As above please improve the final limitations section.

**Opportunities For Improvement:**

-Minor grammar issues: e.g. "InstaNovo have" -> "InstaNovo has", "In the future, We plan to build", "while InstaNovo [3] evaluate the performance on the other data collected by themselves.". Please do a general grammar check throughout the manuscript.

**Relation To Prior Work:**

Yes, the authors did a great job explaining issues with prior datasets, and what steps or features they added that were new in the preprocessing, curation, annotation, model selection, and metrics sections. The figures were also very well done in this part.

**Summary And Contributions:**

The authors have done substantial work in making a comprehensive and useful mass-spectroscopy benchmark for de novo prediction that gets around limitations of prior datasets that were more poorly curated and limited to predefined annotations. There is clear differentiation between old work and their improvements, new useful metrics added to this topic area, and extensive exploration of data quality and feature effects on model performance.

---

> ### Author Rebuttal · Authors · 2024-08-17
>
> *We thank the reviewer for the valuable feedback. We are glad that the reviewer appreciates the contribution of our work. Below, we address the reviewer’s concerns one by one.*
>
>
>
> > **Q1: The limitations section is weak. however, the authors should improve discussion about the weakness of current feature lists, models, data quality or availability, etc. There is basically no limitation stated in the final section though fair limitations are scattered throughout the text in other places.**
>
> A1: Thank you for your suggestions. Given that Data-Independent Acquisition (DIA) data differs from Data-Dependent Acquisition (DDA) data and that there are currently fewer models [1,2] available for DIA, the availability of open-source DIA models is limited. Furthermore, de novo peptide sequencing models have already been widely applied to DDA mass spectrometry data, so our work is currently limited to DDA data. In the future, we will support DIA mass spectrometry data and DIA-specific models.
>
>
>
>
>
> > **Q2: Given how much the MFR degrades model performance, can the authors also improve discussion of how theorists and/or experimentalists are trying to improve that issue in recent work?**
>
> A2:  Thank you for pointing out the impact of the Missing Fragmentation Ratio (MFR) on performance. To address this issue, based on the fact that the sum of the masses of a pair of b and y ions equals the mass of the peptide, and the intensity of a pair of b and y ions is theoretically equal, the model evaluated in the paper, $\pi$-HelixNovo, uses a novel concept of complementary spectra to enhance ion information in the experimental spectrum and improve model performance. Although the original paper does not discuss the impact of MFR on their method, from Figure 2 in our benchmark, we can observe that even with a high MFR ratio, HelixNovo still maintains precision compared to other models, demonstrating that the complementary spectra used by $\pi$-HelixNovo can effectively mitigate the issue of MFR.
>
>
>
>
>
> > **Q3: Minor grammar issues: e.g. "InstaNovo have" -> "InstaNovo has", "In the future, We plan to build", "while InstaNovo [3] evaluate the performance on the other data collected by themselves.". Please do a general grammar check throughout the manuscript.**
>
> A3: Thank you for your valuable feedback regarding the minor grammar issues in our manuscript.  We made the necessary corrections. Specifically:
>
> ​	•	“InstaNovo have” has been corrected to “InstaNovo has.”
>
> ​	•	“In the future, We plan to build” has been revised to “In the future, we plan to build.”
>
> ​	•	The sentence “while InstaNovo [3] evaluate the performance on the other data collected by themselves.” has been revised to "while InstaNovo evaluates its performance on datasets collected by the authors themselves." for clarity and grammatical correctness.
>
> We appreciate your careful review and believe that these changes have improved the overall quality of the manuscript.
>
>
>
> **Reference**
>
> [1]Shiva Ebrahimi and Xuan Guo.  Transformer-based de novo peptide sequencing for data-independent acquisition mass spectrometry. ArXiv:2402.11363
>
> [2] Tran, N.H., Qiao, R., Xin, L. *et al.* Deep learning enables de novo peptide sequencing from data-independent-acquisition mass spectrometry. *Nat Methods* **16**, 63–66 (2019).
>
>
>
>
>
> *We greatly appreciate your helpful comments, as they will undoubtedly help us improve the quality of our article. If our response has successfully addressed your concerns and clarified any ambiguities, we respectfully hope that you consider raising the score. Should you have any further questions or require additional clarification, we would be delighted to engage in further discussion. Once again, we sincerely appreciate your time and effort in reviewing our manuscript. Your feedback has been invaluable in improving our research.*

---

> ### Author Response · Authors · 2024-08-22
>
> Dear Reviewer 9E2x,
>
> We understand that chasing down your reply is not our job and we do not intend to add any pressure on your busy schedule. However, as we are getting closer to the end of the discussion phase, we would really appreciate it if you could be so kind to let us know if we have properly addressed your comments and questions in the rebuttal and if anything can be further clarified.
>
> Many thanks in advance!
>
> Authors

---

> ### Comment · Reviewer_9E2x · 2024-08-22
> **My minor concerns were addressed**
>
> The authors have addressed my short list of minor concerns and I continue to believe it's a technically sound paper worthy of acceptance. However, given that my initial score was fairly high, and in consideration of some of the other shortcomings that the rest of the reviewers pointed out that couldn't be improved without bigger changes to the study design and reporting, I think my initial score remains fair.

---

> > ### Author Response · Authors · 2024-08-23
> > **Thanks very much for the feedbacks!**
> >
> > Dear Reviewer 9E2x,
> >
> > Thank you for your feedbacks! We appreciate your continued support. We will take your comments into account in the revised version.
> >
> > Best regards,
> > Authors

---

### Official Review · Reviewer_ABkg · 2024-07-23
**An interesting benchmark datasets but novelty is limitted**

**Rating:** 6
**Confidence:** 5
**Correctness:** NA

**Review:**

In general, this is an interesting new benchmark. My concern is its novelty as it is a variation of existing benchmark datasets. The novelty part of the work is the additional implementation of the recent methods and evaluation metrics that can unify different works. It could play a role as a reference results for new methods to benchmark against.  I am not confident to give this paper an acceptance but a Marginally above acceptance threshold based on its potential impact for the de novo peptide sequencing research topic.

**Strengths:**

The paper was well written, the implementation of baseline algorithms and he reported results on the standard datasets could be useful for the de novo peptide sequencing research. The analysis on the behavior of the algorithm under the influencing factors such as peptide lengths, noise peaks and missing fragment ratios provide interesting insights might help researchers to find solutions to those problems. The source code and the datasets will be published currently available as an anonymous github repo.

**Additional Feedback:**

NA

**Clarity:**

The paper was very clear and easy to follow, all the analyses are useful and interesting.

**Documentation:**

The documentation in the README of the repo is very brief but containing enough information to reproduce the results in the paper.

**Limitations:**

The limitations have been discussed in the paper but it is too brief with the promises to add more baseline models to  the benchmark.

**Opportunities For Improvement:**

I think on the influencing factors, one additional factor that is important is the difference in the distribution of training/test data, even the data was split by species, there is still a chance that homologous peptides occur across species and thus very hard to evaluate the generalization of the models under homology constraint. An additional analysis with such setting could be useful to evaluate model generalization.

**Relation To Prior Work:**

No particular related prior work from the authors was cited or discussed in the paper.

**Summary And Contributions:**

The paper combines existing De Novo Peptide Sequencing datasets including the Seven-species, Nine-species and the HC-PT to create a new benchmark datasets called the NovoBench. The authors also implemented the most popular machine learning based methods for predicting peptides from spectrum mass data including  DeepNovo, PointNovo, CasaNovo etc  and provide evaluation metrics at amino acid levels, peptide levels and PTM levels. The main claim of the novelty contribution is stated in the abstract: "Firstly, since there is no consensus for the evaluation datasets, the empirical results in different research papers are often not comparable, leading to unfair compari- son. Secondly, the current methods are usually limited to amino acid-level or peptide-level precision and recall metrics. In this work, we present the first unified benchmark NovoBench for de novo peptide sequencing, which comprises diverse mass spectrum data, integrated models, and comprehensive evaluation metrics. Re- cent impressive methods, including DeepNovo, PointNovo, Casanovo, InstaNovo, AdaNovo and π-HelixNovo are integrated into our framework. In addition to amino acid-level and peptide-level precision and recall, we also evaluate the models’ performance in terms of identifying post-tranlational modifications (PTMs), efficiency and robustness to peptide length, noise peaks and missing fragment ratio, which are important influencing factors while seldom be considered. "

---

> ### Author Rebuttal · Authors · 2024-08-17
>
> *We thank the reviewer for the valuable feedback. We are glad that the reviewer appreciates the contribution of our work. Below, we address the reviewer’s concerns one by one.*
>
>
>
> > **Q1: My concern is its novelty as it is a variation of existing benchmark datasets. The novelty part of the work is the additional implementation of the recent methods and evaluation metrics that can unify different works. It could play a role as a reference results for new methods to benchmark against.**
>
> A1:Thank you for your concern regarding the novelty of our paper. We would like to clarify our contributions as follows:
>
> - We unify different models **within a framework for training and evaluation.**
>
> - Unlike previous works that only considered precision and recall at the amino acid and peptide levels, **we are the first to propose more comprehensive metrics, such as Confidence, PTM-level Precision and Recall, etc.**
>
> - We investigate the impact of various factors (peptide length, noise peaks, missing fragments) on model performance, and **the results can provide guidance for users.**
>
> Thank you for point this out and we will highlight this in the revised version.
>
>
>
> > **Q2: I think on the influencing factors, one additional factor that is important is the difference in the distribution of training/test data, even the data was split by species, there is still a chance that homologous peptides occur across species and thus very hard to evaluate the generalization of the models under homology constraint. An additional analysis with such setting could be useful to evaluate model generalization.**
>
> A2:Thank you for your suggestions. We agree with the view that the distribution of the training and test datasets is an important influencing factor. However, we have previously examined the three datasets we used and found that there is no overlap between the peptide sequences in the training set and the test set, making it reasonable to evaluate the model's generalization ability. We will highlight this in the revised version.
>
>
>
>
>
> > **Q3: The limitations have been discussed in the paper but it is too brief with the promises to add more baseline models to the benchmark.**
>
> A3: Thank you for your suggestions. Given that Data-Independent Acquisition (DIA) data differs from Data-Dependent Acquisition (DDA) data and that there are currently fewer models [1,2] available for DIA, the availability of open-source DIA models is limited. Furthermore, de novo peptide sequencing models have already been widely applied to DDA mass spectrometry data, so our work is currently limited to DDA data. In the future, we will support DIA mass spectrometry data and DIA-specific models.
>
>
>
> **Reference**
>
> [1]Shiva Ebrahimi and Xuan Guo.  Transformer-based de novo peptide sequencing for data-independent acquisition mass spectrometry. ArXiv:2402.11363
>
> [2] Tran, N.H., Qiao, R., Xin, L. *et al.* Deep learning enables de novo peptide sequencing from data-independent-acquisition mass spectrometry. *Nat Methods* **16**, 63–66 (2019).
>
>
>
> *We greatly appreciate your helpful comments, as they will undoubtedly help us improve the quality of our article. If our response has successfully addressed your concerns and clarified any ambiguities, we respectfully hope that you consider raising the score. Should you have any further questions or require additional clarification, we would be delighted to engage in further discussion. Once again, we sincerely appreciate your time and effort in reviewing our manuscript. Your feedback has been invaluable in improving our research.*

---

> ### Author Response · Authors · 2024-08-22
>
> Dear Reviewer ABkg,
>
> We understand that chasing down your reply is not our job and we do not intend to add any pressure on your busy schedule. However, as we are getting closer to the end of the discussion phase, we would really appreciate it if you could be so kind to let us know if we have properly addressed your comments and questions in the rebuttal and if anything can be further clarified.
>
> Many thanks in advance!
>
> Authors

---

> ### Author Response · Authors · 2024-08-27
>
> Dear Reviewer ABkg,
>
> Thank you for your valuable feedback on our paper, **such as more detailed explanations of the novelty of our work and the dataset distribution as a influencing factor, and a more detailed description of the limitations. These suggestions have undoubtedly enhanced the quality of our paper.** We understand that chasing down your reply is not our job and we do not intend to add any pressure on your busy schedule. However, as we are getting closer to the end of the discussion phase, we would really appreciate it if you could be so kind to let us know if we have properly addressed your comments and questions in the rebuttal and if anything can be further clarified.
>
> Many thanks in advance!
>
> Authors

---

### Official Review · Reviewer_BQ8s · 2024-07-24
**Comprehensive benchmarking for de novo mass spec peptide sequencing**

**Rating:** 7
**Confidence:** 3
**Correctness:** appears correct
**Clarity:** yes

**Review:**

The overall quality of the work is strong, and the paper is written clearly. The originality of the work is moderate, but is strengthened by its 'extra additions' of evaluating the models across varying data features such as length and fragmentation issues. Novobench could be a very useful tool for the proteomics+ML community going forward as the field of de novo sequencing progresses.

pros:
+fairer evaluation by using consistent datasets
+includes some non-enzymatic PTMs (e.g. oxidized M, etc)
+analyses are broken down using different data features such as peptide length, missing fragments, and efficiency (useful for determining which applications requires what tool).


cons:
-one motivation is to provide guidance for which applications would benefit the most from the different models, but not much discussion or examples are given
-perhaps too much focus on comparing the strengths weaknesses of current methods, and less on how this tool is streamlined for new models to enter the field.
-there is no cross-validation using different one held out test sets, it seems like yeast was always the hold out.
-no inclusion of enzymatic PTMs such as phosphorylation, which are more interesting than non-enzymatic

**Strengths:**

Unclear how much this contribution would spur new researchers into this field, but likely will be useful for those already working in the area to provide fairer comparisons as the SOTA evolves. The code repository appears well documented. No ethical or social implications noted.

**Additional Feedback:**

In DNA sequencing technology, an important metric is accuracy. Does such a metric exist in this application?

**Documentation:**

yes

**Limitations:**

An expanded limitations discussion would be appreciated; as it is now the limitations section mentions no limitations.

**Opportunities For Improvement:**

-Include a greater range of PTMs
-cross validate using additional held out tests to understand the variation of model performance
-build the automated end-to-end pipeline mentioned in the conclusion to increase the impact of the work

**Relation To Prior Work:**

this is discussed adequately.

**Summary And Contributions:**

This work provides a moderately robust set of benchmarks for comparing deep learning models that have been developed for MS-based de novo peptide sequencing. The work compares the most SOTA methods using a wide set of metrics, and additionally compares them under different limitatios (e.g. missing fragments). Both of these features are new contributions to an emerging and important field of research. The code repository is also made freely available and appears well-documented.

---

> ### Author Rebuttal · Authors · 2024-08-17
>
> # Response to Reviewer BQ8s (1/2)
>
> *We thank the reviewer for the valuable feedback. We are glad that the reviewer appreciates the contribution of our work. Below, we address the reviewer’s concerns one by one.*
>
>
>
> > **Q1:  One motivation is to provide guidance for which applications would benefit the most from the different models, but not much discussion or examples are given.**
>
> A1:Thank you for pointing out this issue. Our article can draw several guidance, for instance:
>
> - As observed in Figure 2, $\pi$-HelixNovo, which utilizes complementary spectrum, can effectively mitigate the impact of missing fragments in scenarios with incomplete spectra. This highlights the importance of preprocessing spectra before training, which is universally applicable to other models.
>
> - Additionally, in scenarios involving longer peptide sequences, Transformer-based methods tend to perform better than traditional deep learning methods, demonstrating the superior ability of the transformer architecture to capture long-range dependencies in peptides.
>
> We will include the above discussions and examples in the revised version.
>
>
>
>
>
> > **Q2: Perhaps too much focus on comparing the strengths weaknesses of current methods, and less on how this tool is streamlined for new models to enter the field.**
>
> A2:  Very valuable suggestions! Our benchmark not only evaluates the strengths and weaknesses of current models but also provides the following conveniences for the introduction of new models into this field:
>
> - **Unified Datasets and Interfaces for Fair Comparison**: We have selected and preprocessed a variety of datasets for the de novo peptide sequencing task, and standardized them into a consistent dataset class for easy use.
> - **Consistent and Diverse Evaluation Metrics**: Within the same code framework, we have implemented various evaluation metrics, including those considered in previous work as well as new metrics proposed by us.
> - **Convenient Baseline Comparison**: The implementation frameworks of previous methods differ significantly; we have integrated these different approaches to enable faster comparison of new methods with existing ones.
> - **Evaluation under Different Influencing Factors:** Our benchmark can assess the performance of new models under various conditions such as peptide length, noise peaks, and Missing Fragmentation Ratio (MFR). This enables a more in-depth exploration of new models and facilitates better iteration.
>
> In summary, once a new model is implemented, our benchmark allows for convenient evaluation of its performance across different datasets. Thank you for your suggestions and we will include them in the revised version.
>
>
>
> > **Q3: There is no cross-validation using different one held out test sets, it seems like yeast was always the hold out.**
>
> A3:  Thank you for your suggestion. Given the high computational cost (e.g., training CasaNovo for 30 epochs on a nine-species dataset requires 40 machine hours on an NVIDIA A100 80G), we selected CasaNovo and AdaNovo as the models for comparison. We conducted experiments using the nine-species dataset, excluding human and honeybee. Table Re 1 shows the performance of CasaNovo and AdaNovo across different holdout datasets.
>
>
>
> Table Re1: The performance of casanovo and adanovo on different hold out species datasets.
>
> | Method   | Dataset                       | aa precision | aa recall | pep precision | ptm recall | ptm precision | Auc   |
> | -------- | ----------------------------- | ------------ | --------- | ------------- | ---------- | ------------- | ----- |
> | CasaNovo | Nine species exclude human    | 0.549        | 0.550     | 0.332         | 0.426      | 0.555         | 0.285 |
> | AdaNovo  | Nine species exclude human    | 0.571        | 0.571     | **0.337**     | 0.440      | 0.530         | 0.298 |
> | CasaNovo | Nine species exclude honeybee | 0.774        | 0.773     | 0.579         | 0.626      | 0.771         | 0.546 |
> | AdaNovo  | Nine species exclude honeybee | 0.772        | 0.771     | **0.585**     | 0.638      | 0.731         | 0.554 |
>
>
>
> The primary metric for assessing the model's practical utility is peptide-level precision, as the objective is to accurately assign a complete peptide sequence to each spectrum. From Table 1, we can observe that AdaNovo consistently outperforms CasaNovo across different holdout species datasets. **More importantly, our framework allows users to evaluate model performance using a wide variety of  datasets. Additionally, we provide the HC-PT dataset, beyond the cross-validation dataset, for more comprehensive evaluation.**

---

> > ### Author Rebuttal · Authors · 2024-08-17
> >
> > # Response to Reviewer BQ8s (2/2)
> >
> > > **Q4: No inclusion of enzymatic PTMs such as phosphorylation, which are more interesting than non-enzymatic**
> >
> > A4:  Excellent suggestion! We selected Casanovo and Adanovo as the models for comparison and conducted experiments on datasets containing Ymod phosphorylation  and  Kmod Acetylation and Kmod Biotinylation. Table Re 2 summarizes the dataset statistics, while Table Re 3 presents the performance of Casanovo and Adanovo on datasets with different modifications.
> >
> >
> >
> > Table Re2: The datasets statistics.
> >
> > | Dataset                 | precusor m/z | precusor charge | Peaks num. | Intensity | Peptide len. | PTM   class | min m/z | max m/z | Train/valid/test num. |
> > | ----------------------- | ------------ | --------------- | ---------- | --------- | ------------ | ----------- | ------- | ------- | --------------------- |
> > | Ymod phosphoryl dataset | 654.90       | 2.13            | 390.16     | 282347.13 | 11.24        | 2           | 100.00  | 1999.88 | 32.367/4,045/4,047    |
> > | Kmod Acetyl dataset     | 660.97       | 2.12            | 234.97     | 111970.51 | 12.24        | 2           | 99.99   | 1996.66 | 40,362/5,079/5,079    |
> > | Kmod Biotinyl dataset   | 649.49       | 2.15            | 227.67     | 123996.81 | 11.44        | 2           | 100.00  | 1999.99 | 41,684/5,210/5,211    |
> >
> >
> >
> >
> >
> > Table Re 3: The performance of casanovo and adanovo on different modifications datasets.
> >
> > | Method   | Dataset         | aa precision | aa recall | pep precision | ptm recall | ptm precision | Auc    |
> > | -------- | --------------- | ------------ | --------- | ------------- | ---------- | ------------- | ------ |
> > | CasaNovo | Ymod phosphoryl | 0.9954       | 0.9954    | 0.9938        | 0.9931     | 0.9931        | 0.9934 |
> > | AdaNovo  | Ymod phosphoryl | 0.9959       | 0.9959    | **0.9940**    | 0.9945     | 0.9945        | 0.9937 |
> > | CasaNovo | Kmod Acetyl     | 0.9942       | 0.9942    | **0.9931**    | 0.9985     | 0.9985        | 0.9928 |
> > | AdaNovo  | Kmod Biotinyl   | 0.9937       | 0.9941    | 0.9929        | 0.9983     | 0.9971        | 0.9926 |
> > | CasaNovo | Kmod Acetyl     | 0.9936       | 0.9948    | **0.9925**    | 0.9887     | 0.9874        | 0.9921 |
> > | AdaNovo  | Kmod Biotinyl   | 0.9472       | 0.9426    | 0.9387        | 0.8689     | 0.8852        | 0.9369 |
> >
> >
> >
> > From Table Re 3, we can observe that AdaNovo does not consistently outperform CasaNovo across most datasets with modifications, likely due to its sensitivity to adaptive training hyperparameters. We will include these results in the revised version. Thank you for your valuable suggestion and we believe that these changes have improved the overall quality of the manuscript.
> >
> >
> >
> > > **Q5: An expanded limitations discussion would be appreciated; as it is now the limitations section mentions no limitations.**
> >
> > A5: Thank you for your suggestions. Given that Data-Independent Acquisition (DIA) data differs from Data-Dependent Acquisition (DDA) data and that there are currently fewer models [1,2] available for DIA, the availability of open-source DIA models is limited. Furthermore, de novo peptide sequencing models have already been widely applied to DDA mass spectrometry data, so our work is currently limited to DDA data. In the future, we will support DIA mass spectrometry data and DIA-specific models.
> >
> >
> >
> > > **Q6: In DNA sequencing technology, an important metric is accuracy. Does such a metric exist in this application?**
> >
> > A6: In this application, accuracy is not a relevant metric. For the task of de novo peptide sequencing, we consider amino acid-level precision, recall, and peptide-level precision. Among these metrics, the primary metric for assessing the model's practical utility is peptide-level precision, as the objective is to accurately assign a complete peptide sequence to each spectrum.
> >
> >
> >
> >
> >
> > **Reference**
> >
> > [1]Shiva Ebrahimi and Xuan Guo.  Transformer-based de novo peptide sequencing for data-independent acquisition mass spectrometry. ArXiv:2402.11363
> >
> > [2] Tran, N.H., Qiao, R., Xin, L. *et al.* Deep learning enables de novo peptide sequencing from data-independent-acquisition mass spectrometry. *Nat Methods* **16**, 63–66 (2019).
> >
> >
> >
> >
> >
> > *We greatly appreciate your helpful comments, as they will undoubtedly help us improve the quality of our article. If our response has successfully addressed your concerns and clarified any ambiguities, we respectfully hope that you consider raising the score. Should you have any further questions or require additional clarification, we would be delighted to engage in further discussion. Once again, we sincerely appreciate your time and effort in reviewing our manuscript. Your feedback has been invaluable in improving our research.*

---

> ### Author Response · Authors · 2024-08-22
>
> Dear Reviewer BQ8s,
>
> We understand that chasing down your reply is not our job and we do not intend to add any pressure on your busy schedule. However, as we are getting closer to the end of the discussion phase, we would really appreciate it if you could be so kind to let us know if we have properly addressed your comments and questions in the rebuttal and if anything can be further clarified.
>
> Many thanks in advance!
>
> Authors

---

> > ### Comment · Reviewer_BQ8s · 2024-08-23
> >
> > The authors have thoroughly addressed the main points of my concerns and my support for accept as a good paper stands.

---

> > > ### Author Response · Authors · 2024-08-24
> > > **Thanks very much for the insightful and helpful reviews!**
> > >
> > > Dear Reviewer BQ8s,
> > >
> > > Thank you for your positive feedback and continued support for our paper! We appreciate your thoughtful review and are glad that we have adequately addressed your concerns.
> > >
> > > Best regards,
> > > Authors.

---

### Official Review · Reviewer_c3Hs · 2024-08-02
**Careful integration of multiple key proteomics workflows**

**Rating:** 6
**Confidence:** 3
**Correctness:** Yes
**Clarity:** Reasonably so.

**Review:**

The authors have identified the bottleneck challenges in benchmarking peptide sequencing methods and have taken pains to ensure that state of the art methods can operate within their benchmarking framework. Authors have defined a number of reasonable metrics for evaluation, as well as identifying common challenges in the real-world problem (e.g., noise; fragmentation information) that the benchmark ought to reflect.

**Strengths:**

I think the paper's careful examination of state of the art models, including computational efficiency, is a real strength! The code is available with clear instructions for installation, and the topic in question is a valuable experimental process that needs benchmarks.

**Additional Feedback:**

n/a

**Documentation:**

Code and data are both available and reasonably well documented

**Limitations:**

The authors' discussion of limitations is pretty cursory: "In the future, our benchmark will support more models and datasets." Describing limitations only in terms of scope is a reasonable plan but it's low-hanging fruit. What are the *scientific* limitations?

**Opportunities For Improvement:**

There aren't significant opportunities for improvement without appreciably altering the scope of the work

**Relation To Prior Work:**

There aren't previous such benchmarks, at least per the work.

**Summary And Contributions:**

Authors propose Novobench, a benchmark for peptide identification by mass spec proteomics, and evaluate prior art on that benchmark.

---

> ### Author Rebuttal · Authors · 2024-08-17
>
> *We thank the reviewer for the valuable feedback. We are glad that the reviewer appreciates the contribution of our work. Below, we address the reviewer’s concerns one by one.*
>
>
>
> > **Q1: The authors' discussion of limitations is pretty cursory: "In the future, our benchmark will support more models and datasets." Describing limitations only in terms of scope is a reasonable plan but it's low-hanging fruit. What are the *scientific* limitations?**
>
> A1: Thank you for your suggestions. Given that Data-Independent Acquisition (DIA) data differs from Data-Dependent Acquisition (DDA) data and that there are currently fewer models [1,2] available for DIA, the availability of open-source DIA models is limited. Furthermore, de novo peptide sequencing models have already been widely applied to DDA mass spectrometry data, so our work is currently limited to DDA data. In the future, we will support DIA mass spectrometry data and DIA-specific models.
>
>
>
> **Reference**
>
> [1]Shiva Ebrahimi and Xuan Guo.  Transformer-based de novo peptide sequencing for data-independent acquisition mass spectrometry. ArXiv:2402.11363
>
> [2] Tran, N.H., Qiao, R., Xin, L. *et al.* Deep learning enables de novo peptide sequencing from data-independent-acquisition mass spectrometry. *Nat Methods* **16**, 63–66 (2019).
>
>
>
>
>
> *We greatly appreciate your helpful comments, as they will undoubtedly help us improve the quality of our article. If our response has successfully addressed your concerns and clarified any ambiguities, we respectfully hope that you consider raising the score. Should you have any further questions or require additional clarification, we would be delighted to engage in further discussion. Once again, we sincerely appreciate your time and effort in reviewing our manuscript. Your feedback has been invaluable in improving our research.*

---

> ### Author Response · Authors · 2024-08-22
>
> Dear Reviewer c3Hs,
>
> We understand that chasing down your reply is not our job and we do not intend to add any pressure on your busy schedule. However, as we are getting closer to the end of the discussion phase, we would really appreciate it if you could be so kind to let us know if we have properly addressed your comments and questions in the rebuttal and if anything can be further clarified.
>
> Many thanks in advance!
>
> Authors

---

> ### Author Response · Authors · 2024-08-27
>
> Dear Reviewer c3Hs,
>
> Thank you for your valuable feedback on our paper, such as **increase the explanation of scientific limitations. These suggestions have undoubtedly enhanced the quality of our paper.** We understand that chasing down your reply is not our job and we do not intend to add any pressure on your busy schedule. However, as we are getting closer to the end of the discussion phase, we would really appreciate it if you could be so kind to let us know if we have properly addressed your comments and questions in the rebuttal and if anything can be further clarified.
>
> Many thanks in advance!
>
> Authors

---

### Author Rebuttal · Authors · 2024-08-17

# General Response

We thank the reviewers for their insightful and constructive feedback on our manuscript. We are encouraged by their recognition that **our work is strong and the paper is well-written (Reviewers BQ8s, ABkg, 9E2x)**. We also appreciate their acknowledgment that **the new metrics and influencing factors we proposed are both inspiring and important** to the field, helping researchers uncover new insights **(Reviewers c3Hs, BQ8s, ABkg, 9E2x)**. The reviewers highlighted that **the topic of our benchmark is significant and interesting (Reviewers c3Hs, BQ8s, ABkg, 9E2x)** and recognized that our work has produced **a comprehensive and robust benchmark (Reviewers c3Hs, BQ8s, ABkg, 9E2x)**. Additionally, we appreciate the recognition that **the code is available and reasonably well-documented (Reviewers c3Hs, BQ8s, ABkg, 9E2x)**.

Based on the reviewers’ valuable feedback, we have conducted several additional experiments. Below, we address the issues pointed out by the reviewers and resolve any possible misunderstandings:

**1.Clarifications:**

- **Limitations and future work:** We have revised the limitations and future work sections.
- **The novelty of the benchmark**: Our benchmark greatly facilitates the entry of new models into this field. It provides a unified framework for training and testing, introduces new metrics, and identifies influencing factors that address key challenges in the field.
- **Dataset distribution**: We clarify that there is no data overlap in the datasets we used.
- **Insights provided by the benchmark**: We discussed the insights that can be drawn from the results of this benchmark.

**2.New Experiment Results:**

- **Results across different hold out species**: We conducted tests on datasets where different species were held out.
- **Diverse PTMs**: We conducted tests on various post-translational modifications (PTMs).

**3.Minor Issues:** We have corrected the existing grammatical issues.



***Thank you for your time and effort in reviewing our manuscript. We sincerely appreciate your insightful comments, which will certainly enhance the quality of our article. If our response has effectively addressed your concerns and resolved any ambiguities, we respectfully hope that you consider raising the score. Should you have any further questions or need additional clarification, we would be delight to engage in further discussion. Once again, we sincerely appreciate your time and effort in reviewing our manuscript. Your feedback has been invaluable in improving our research.***

---

### Decision · Program_Chairs · 2024-09-26

**Decision:**

Accept (Poster)

**Comment:**

This paper unifies disparate datasets and benchmarks for de novo peptide sequencing using mass spec. Although no new data is presented, curating the existing datasets, providing a unified benchmark, and retraining and evaluating existing models against it in order to make head-to-head comparisons of what works and does not work is a major contribution.

Strengths:
- Important problem with a lot of disparate methods and datasets (significance)
- Very nice, concise, summary of the de novo peptide sequencing problem (clarity)
- A unified dataset evaluated on many recent models with thorough metrics (many new) allows insights into what works and what still needs research effort (quality, originality)

Weaknesses:
- A better discussion of the limitations would be nice (clarity, significance)
- No new data